# The Bacterial Urban Resistome: Recent Advances

**DOI:** 10.3390/antibiotics11040512

**Published:** 2022-04-12

**Authors:** Alberto Vassallo, Steve Kett, Diane Purchase, Massimiliano Marvasi

**Affiliations:** 1School of Biosciences and Veterinary Medicine, University of Camerino, 62032 Camerino, Italy; alberto.vassallo@unicam.it; 2Department of Natural Sciences, Middlesex University London, London NW4 4BT, UK; s.kett@mdx.ac.uk (S.K.); d.purchase@mdx.ac.uk (D.P.); 3Department of Biology, University of Florence, 50019 Sesto Fiorentino, Italy

**Keywords:** antimicrobial resistances, urban microbiome, hospitals, pet, recreational water, ARB, ARM, ARGs

## Abstract

Cities that are densely populated are reservoirs of antibiotic resistant genes (ARGs). The overall presence of all resistance genes in a specific environment is defined as a resistome. Spatial proximity of surfaces and different hygienic conditions leads to the transfer of antibiotic resistant bacteria (ARB) within urban environments. Built environments, public transportation, green spaces, and citizens’ behaviors all support persistence and transfer of antimicrobial resistances (AMR). Various unique aspects of urban settings that promote spread and resilience of ARGs/ARB are discussed: (i) the role of hospitals and recreational parks as reservoirs; (ii) private and public transportation as carriers of ARGs/ARB; (iii) the role of built environments as a hub for horizontal gene transfer even though they support lower microbial biodiversity than outdoor environments; (iv) the need to employ ecological and evolutionary concepts, such as modeling the fate of a specific ARG/ARB, to gain enhanced health risk assessments. Our understanding and our ability to control the rise of AMR in an urban setting is linked to our knowledge of the network connecting urban reservoirs and the environment.

## 1. Introduction

More than half of the world’s population lives in urban areas and, increasingly, within high population-density cities [1,2]. Some built environments undergo frequent and high-volume human throughput/activity and influenced strongly by their surrounding environments. These may be considered ‘unrestricted’ buildings (e.g., offices, retail centres, schools, and farms). Other buildings are more ‘restricted’ in that they permit only limited access and within which operations aimed at reducing/eliminating microbes and/or frequent cleaning take place (e.g., clean-room facilities and intensive care units). Cities also include outdoor environments (recreational parks, ponds, lakes) that can support high human throughput and activity. Such environments, either in the same city or in different ones, are connected via public transport (e.g., metro, buses, air transport), and shared transport (e.g., cars, scooters, bicycles). Waterborne linkage between and within these environments occurs via runoff over impermeable surfaces and wastewater transport within closed and open drainage and sewerage systems. Additional linkages occur via airborne movement of microbes or of materials colonised by them.

One consequence of high human densities within this complex array of built environments, associated outdoor environments, and their infrastructural and human transport links, is that there is substantial opportunity for the transport of antibiotic resistant bacteria (ARB) and the antibiotic resistance genes (ARGs) they contain. The sum of all such genes conferring antibiotic resistance is defined as a ‘resistome’, and it includes those present in both pathogenic and non-pathogenic microorganisms. 

The resistome includes genes commonly present in nature [3,4], evolved through antibiotic selective pressure, novel (anthropogenic) resistance genes emerged from proto-resistance genes, and cryptic genes that are already present but not expressed [3]. All this material is multiplied and boosted within the microbial population through mutations, horizontal gene transfer, and mobile genetic elements for transfer between environments [5]. 

The urban resistome (along with ARB) undergoes intense selection accompanied by continual export and import via biotic and abiotic vectors (e.g., aerosols, water courses, and water bodies) [6,7,8,9,10]. This ensures that urban inter- and intra-resistome ARGs’ flux is high and that antimicrobial resistance genes spread rapidly and widely within cities [11]. Each city possesses its own microbial signature within which specific ARGs demonstrate different distributions determined by, for example, temperature, surface materials, elevation, proximity to the coast, population density, and geographical region [11] (Figure 1A). 

Confined environments, where continuous cleaning and/or confined habitat select specific ARGs/ARB, tend to support so-called ‘man-made microbial resistances’ differing from those observed in open environments. Indeed, antibiotic resistance mechanisms can be grouped according to the environment in which they occur, such as ‘microbial confinement’ versus ‘open’ environments (Figure 1B) [12].

Both ARGs and ARB can be considered as contaminants of emerging concern (CEC) and, in particular, they belong to the recently proposed category of *evolving CEC* (e-CEC) [13,14]. Currently, hundreds of antibiotic resistance genes are known [15], and, contrary to standard CEC molecules, e-CEC can evolve, duplicate and escape human control. For this reason, ARGs require particularly attention and deviate from regular risk assessment [13]. 

This review gives a general overview of urban ARGs/ARB diversity, the resistomes they comprise and functional interconnections within urban environments [16,17].

## 2. Built Environments

People from developed countries spend about 90% of their time indoors [18,19]. Such environments are extremely heterogeneous in composition, experience widely different air circulation patterns and may harbour dust or other organic detritus. 

Humans and pets contribute to the microbial community of these environments, as each individual can transfer its microbial fingerprint to indoor spaces [20]. Thus, because of the varied composition and activities within, built environments are potential hotspots for microbial exchange and spread [21,22] (Figure 2).

The resistome of built environment is largely focused upon hospitals. Hospitals are a central hub for the spread of resistances, and the relevant literature is substantial (Figure 3A).

To attempt a comprehensive review of AMR in hospitals would diverge from the purposes of this paper. Because, however, literature shows hospitals are reservoirs and incubators for new ARG variants [23], we performed a systematic research of the frequency of reports regarding the top 50 ARG subtypes in hospitals (and their respective antibiotic families) as described by Zhuang et al. [17], to determine their role as sources for urban resistomes. Data extracted from PubMed publications from 1990 to the present (Appendix A) showed that the families of β-lactams and *mecA* genes (involved in methicillin resistance) [24] were the most reported. Potential escape routes from hospitals include wastewater effluents, biohazard operators, patients, and visitors [25]. Another study showed that even dust from indoor hospital samples exhibited a complex resistome profile, with an average ARG concentration of 0.00042 copies/16S rRNA gene. This study also found that the outpatient hall was one of the main ARG transmission sources, permitting distribution of ARGs to other departments [26]. Among ARGs identified, this work reported: *aadE, ARRAM, mecA*, *aad(9)*, *brcA*, *tetL*, *InuB*, *tet40*, *tetZ*, *tetA*, *tetK*, *norA*, *CE*, *aadD*, *qacA*, *vgaA*, *tetK*, *tetH*, *tetG*, *mexT*, *OXA*, *aph(3′)*, *mefA*, *bleO*, and *CfxA2*. Hospital air-conditioners can distribute the resistome over time and accumulate ARGs whilst transporting them within departments [26]. Genes present in dust were also shown to change with season: for example, 86 ARGs subtypes were detected in winter, whilst only 11 occurred in summer.

Cleaning chemicals can also affect microbial communities and associations between antimicrobials and bacterial resistomes [27]. Increased confinement and cleaning are associated with a loss of microbial diversity [12]. In this context, Mahnert et al. [12] compared ‘human-driven’ microbial resistomes present on surfaces in clinical settings and in other built environments. Their metagenomic approach showed that environments with increased confinement and cleaning were associated with genomes enriched with functions related to virulence, disease, defence, stress response, and resistance against five classes of antibiotics (Figure 1B). In contrast, unrestricted buildings were characterized by a higher diversity of bacteria associated with the outdoor environment and processed food [12]. Confinement and cleaning were associated with a shift from Gram-positive bacteria, such as Actinobacteria and Firmicutes, to Gram-negative, such as Proteobacteria and that the loss of microbial diversity correlated with an increase in resistance [12]. 

Other examples showed the distribution of ARGs on floor surfaces in different types of environments. Gupta et al. (2019) sampled the carpet and vinyl floors from medical, veterinary, and office buildings in both high- and low-traffic areas [28]. Results showed a widespread distribution of *tetQ* and *sul1* resistant genes in all sample areas, while carbapenemase encoding genes from *Klebsiella pneumoniae* were only detected from the high-traffic surfaces of medical facilities. Most indoor environments harboured ARGs, veterinary samples had higher concentrations of *tetQ* and *sul1*, and carbapenem resistance was only observed in the medical centre. Moreover, most floor surfaces also showed the presence of dog- and bird-specific faecal bacteria. 

Another less commonly considered aspect is the “thanato-resistome” associated with locations and practices related to the funeral industry. Gwenzi [29] suggested that all such environments should be considered as potential reservoirs of ARGs and may present health risks for funeral workers. It is likely that human cadavers harbour resistant microbes and/or ARGs so, consequently, all wastes derived from thanatopraxy (e.g., embalming fluids) should also be considered as potentially contaminated. In addition, decomposition of buried human bodies and discharge of wastewater from thanatopraxy care facilities may eventually contribute to the spreading of resistance in the urban environment [29].

Schools are environments of particular interest, since direct (i.e., human-to-human) and indirect (i.e., object-mediated) contact derived transmissions of bacteria are common. Although schools constitute high-risk environment for infections, literature regarding metagenome shotgun analysis of resistomes in this environment is missing. More investigations are needed.

## 3. Transportation

### 3.1. Air Transport

Airports and aeroplanes are a major causal influence upon ARGs and ARB dispersal. Every year billions of passengers are transported between countries and continents. ARGs have been detected in airport wastewater treatment plants (WWTP). The role of aeroplanes, and particularly their toilet sewage, as a source of ARGs has been investigated. Two studies agreed that aeroplane-borne sewage can effectively contribute to the fast and global spread of antibiotic resistance [30,31]. 

Shotgun sequencing of toilet waste from 18 international aeroplane flights arriving in Copenhagen, Denmark, from nine cities and three different world regions showed that 0.06% of all DNA sequencing reads were assigned to resistance genes and that the most abundant genes encoded resistance against tetracyclines, macrolides, and β-lactams. The relative abundance of *sul1* (sulfonamide) and *tetM* (tetracycline) resistance genes were significantly increased in aeroplane samples compared to the airport’s WWTP inlets. Median relative abundances between the two sample origins (aeroplane sewage versus airport’s WWTP) differed by factors of 5 (*sul1*) and 18 (*tetM*), respectively. Flights from South Asia showed significantly higher abundance and diversity of *bla*_CTX-M_ genes compared to those from North America. Detection of antibiotic resistances was also associated with the presence of *Salmonella enterica* (higher from South Asia), and *Clostridium difficile* in samples from North America [31]. However, functional-taxonomical tests were not performed; therefore, it is not possible to associate resistances with specific pathogens. When, however, aeroplane-borne sewage was tested for ARGs in terms of diversity and quantity, resistances against fluorochinolones, third-generation cephalosporins, and aminoglycosides were particularly associated with *Escherichia coli* isolated from the sewage [30]. Comparison with municipal sewage also showed that the aeroplane sewage had more mobile ARG elements, with higher relative abundances. The study [30] also stressed that ARG-concentrated aeroplane sewage is discharged into airport WWTPs so both WWTP influents and effluents should be monitored for their ARG/ARB profiles.

Screening of military aviators showed the presence of methicillin-resistant *Staphylococcus aureus* (MRSA). MRSA infections are significantly more frequent among members of the military than in the general population and some types of MRSA related to military personnel evolves separately from nosocomially acquired MRSA [32]. Attention was raised because, community-acquired, non-healthcare associated MRSA-based resistance might be transferred to dense urban populations [33].

### 3.2. Resistomes in Trains and Metro

Of all public transport types worldwide, trains and metro are probably those with the highest number of passengers. In the largest cities, several million people use these transport methods daily. Because of confined spaces, poor air circulation and prolonged skin-surface contact within coaches (e.g., handrails), this kind of public transport influences passengers’ skin microbiota [34], contributing to the diffusion of resistant strains as well [35,36]. Like aeroplanes, trains also transport people long distances, between urban areas and across international borders. 

Resistance patterns of airborne bacteria in the Shanghai (China) metro were analysed finding *Staphylococcus* strains carrying *mecA* and *qac* resistance genes, which confer resistance against methicillin and lactams antibiotics, respectively [37]. Frequencies were compared with those from hospital samples and from control samples taken from parks. Frequencies of detection of resistance genes in metro and hospital samples were comparable but both were higher than in park samples. Similar results were also obtained in two studies regarding several surfaces types in railway stations in Guangzhou (China) [38,39], where about 75% of *Staphylococcus* isolates were from multidrug resistant (MDR) strains.

As with aeroplane transport, wastewater produced in trains can contribute to the spreading of resistance genes. Wei and colleagues [40] investigated the efficiency of a pilot-scale system in removing ARGs and ARB in wastewater collected from multiple units of high-speed trains in Beijing (China). They showed that the abundance of ARGs and mobile genetic elements was similar to that of untreated hospital wastewater and higher than that of domestic wastewater.

### 3.3. Shared Transportation

Use of shared transport is increasing in popularity, especially in larger cities. Citizens can both save money and reduce overall emissions of greenhouse gases by sharing bicycles, cars, and scooters. Sharing vehicles, however, increases the risk of transferring ARGs by indirect host-to-host transfer. This risk occurs via prolonged contact with surfaces, such as handles and seats. In 2019 in Chengdu (China), there were 1.23 million shared bicycles used for more than 2 million daily rides. Resistant Gram-positive bacteria were isolated from bicycles or riders, and hosted resistances against clinically important antibiotics including linezolid, fosfomycin, and vancomycin, with a significant quantity of these isolates showing multidrug resistance. MRSA strains were also isolated and whole genome sequencing further detected the presence of *fosB*, *fusB*, and *lnu*(G) in *S. aureus* and *optrA* in enterococci, in addition to other genes. Bacterial transmission across geographical-distinct locations (both bicycles and riders) was demonstrated by genetically closely-related bacteria [41]. 

Another study aimed to address the risk of public shared bicycles transferring resistant strains of *Staphylococcus epidermidis* within a population. Antimicrobial susceptibility and molecular testing were performed to classify the *Staphylococcus* species, resistance patterns, presence of *mecA* gene, and clonal lineage. Overall, 49% of screened staphylococci were *mecA*-positive with a high diversity of staphylococcal cassette chromosome *mec* (*SCCmec*) elements [42]. Such variability of *SCCmec* could be associated with a high variability within *Staphylococcus* species, showing a strong propensity to dissemination [42].

Another bacterial family commonly isolated from shared bicycles are the Enterobacteriaceae. Sampling 2117 shared bicycles at 240 metro stations in Beijing showed a total of 444 non-duplicate Enterobacteriaceae isolated from 418 samples at 166 stations. In this case, the isolates were resistant to amikacin (0.7%), ceftazidime (0.7%), ciprofloxacin (0.5%), colistin (3.6%), doxycycline (5.4%), gentamicin (1.3%), florfenicol (2.5%), fosfomycin (6.3%), and meropenem (0.5%). Moreover, 31.5% were resistant to sulfamethoxazole-trimethoprim. Three ceftazidime-resistant *E. coli* isolates were positive for *bla*_CTX-M-199_ and two were positive for carbapenemase-producing gene *bla*_NDM-5_ [43]. In another study investigating hand-bicycles in China, *Bacillus* spp. Were also found to be resistant to bacitracin and sulfamethoxazole [44].

Multivariate logistic regression of data regarding resistant bacteria sampled from shared bicycles at metro stations near hospitals revealed that variable ‘secondary/tertiary non-profit hospital nearby’ was significantly (*p* < 0.05) associated with isolation of Enterobacteriaceae from the shared bicycles [43]. This indicates that these resistances may have a common origin in hospitals [43].

Such data suggest the use of shared bicycles increases risk of ARGs/ARB dissemination and, equally, suggest the need for an effective disinfection strategy.

## 4. Urban Green Spaces and Parks

Green spaces provide important ecosystem services and improve citizens’ physical and mental well-being and development [45,46,47,48]. Hence, most epidemiological studies of urban green exposure focus on their availability and health benefits, few studies examined unwanted side effects, such as pollen allergies, vector-borne diseases [49], or their role as potential reservoirs of antibiotic resistome.

Soil samples taken anywhere on the planet contain antibiotic resistant microorganisms (ARMs), therefore this is also true for soil sample obtained from urban parks. It has been reported that urban park soil microbiomes are both rich in biosynthetic diversity and distinct from non-urban samples in their biosynthetic gene composition [50]. Anthropogenic ARGs/ARM can enter these urban ecosystems via several pathways: faecal shedding by animals (e.g., domestic dogs and wild animals, especially mammals and birds), irrigation with reclaimed water and atmospheric deposition.

Such diversity is further encouraged via horizontal gene transfer (HGT) mediated by mobile genetic elements that facilitate the transfer of ARGs within and between related and unrelated bacterial species. As a consequence, such ARGs have the potential to become widespread within microbial communities in domestic and feral animal populations. Worsley-Tonks et al. [51] reported that faecal samples and rectal swabs of stray dogs had twice as many unique ARGs compared to owned dog samples, which was partly driven by a greater richness of beta-lactamase genes conferring resistance to penicillin and cephalosporin. Other urban wild animals, such as foxes, are more likely to be exposed to AMR bacteria and resistance drivers from food waste, garbage, sewage, wastewater, and consumption of contaminated prey than those living in remote areas. Mo et al. [52] found that the total occurrence of AMR in *E. coli* from faecal swabs of foxes in areas with high population density was significantly high. 

Wild birds, such as geese, swans, and gulls, are frequent visitors to many urban parks, particularly those with ponds and lakes. Many migratory birds also come into close contact with humans in urban areas where they feed on terrestrial grasses found in public parks and sports grounds. These wild birds may also play a role in transporting antibiotic resistance to urban green spaces via faecal shedding [53,54,55,56]. 

ARGs in recreational urban water bodies are also an issue. Urban ponds are utilised in a variety of ways; some are used for recreation, others receive flood relief water from rivers or store urban runoff (Figure 3B,C) [24]. Sewage leaking into recreational water is a serious issue, second only to hospitals in terms of ARGs diversity (Figure 3B). With an average absolute abundance of 1.38 × 10^7^ copies/mL ARGs and 4.19 × 10^6^ copies/mL mobile genetic elements, urban ponds can be considered as ARGs hotspots [10] (Figure 3C).

Water scarcity is increasingly a challenge for industrial and urban development, especially in arid and semi-arid regions. To ensure a sustainable water supply, water-reuse and water management concepts have been proposed by a number of researchers requiring the integration of grey infrastructures (water supply and wastewater treatment) with green infrastructures, such as parks and public green spaces [57,58]. For example, in Australia during the period 2009–2010, average state-wide use of recycled water for urban irrigation was 27.2% whilst the nation-wide average of total recycled water produced was 14%. In Madrid, Spain, irrigating urban park turf with reclaimed water has led to grass biomass increase [59] and, in terms of micronutrient content, the reclaimed water used was adequate for irrigation [60].

Reclaimed water may contain ARB that could be transferred to the urban environment via irrigation. Limayem et al. [61] detected the presence of drug resistance in both pathogenic and non-pathogenic bacterial strains in reclaimed water samples, where isolated *Escherichia*, *Klebsiella,* and *Acinetobacter* displayed resistance to chloramphenicol, ciprofloxacin, daptomycin, erythromycin, gentamycin, kanamycin, streptomycin, lincomycin, linezolid, nitrofurantoin, penicillin, quinupristin/dalfopristin, tertacycline, tigecycline, tylosin tartrate, and vancomycin. *Pseudomonas* was resistant to ciprofloxacin, erythromycin, daptomycin, lincomycin, linezolid, nitrofurantoin, and tigecycline. Moreover, *Streptococcus* and *Staphylococcus* were resistant to daptomycin, kanamycin, lincomycin, linezolid, nitrofurantoin, penicillin, quinupristin/dalfopristin, tylosin tartrate, and vancomycin [61]. 

The irrigation of urban parks with treated wastewater significantly increased the abundance and diversity of various antibiotic resistance genes (resistances to β-lactam were the most prevalent ARG type), although significant increase in horizontal gene transfer was not observed [62]. The potential for such transfers exists, however, Yan et al. [63] reported that diverse ARGs and mobile genetic elements, including six transposon-transposase genes, class 1 integron genes (intI1 and cintI1) were present in both urban park grass phyllosphere and soil. Such genes indicated the potential for horizontal gene transfer of soil ARGs.

These cases demonstrate that although there is increasing interest in the use of tertiary wastewater from WWTP for various applications, primarily agricultural and landscape irrigation, there may be very real risks associated with such uses in terms of enhanced resistome profiles within recipient environments.

Atmospheric deposition of industrial pollutants may also play an important role in shaping ARG profiles. In a study of ARGs abundance in bioaerosol and particulate matter (PM_2.5_) under different rain conditions, ARGs were detected in 8 out of 21 rain events [64], suggesting that wet and dry deposition could contribute to urban green space/park resistomes. Further studies should be focused upon differentiating between normal, natural resistance patterns, and ARGs that may have been introduced from other urban routes (such as hospitals).

## 5. Companion Animals

The potential risk of transmitting antimicrobial resistant isolates from animals to humans and vice versa is associated with close contact between animals and their guardians. This is especially true in urban environments, where companion animals share close proximity with humans within the household. The frequency of publications on ARGs with reference to pet and companion animals is similar to that of recreational water (Figure 3D). Pets may act as a reservoir for self-infection, further transmission to other hosts, and to the environment. For example, extended-spectrum cephalosporin (ESC)-resistant Enterobacteriaceae clonal spread has been observed among companion animals and between companion animals and the environment [65]. The spread of extended-spectrum-cephalosporinases can be due to successful combination of particular ESBL/AmpC encoding genes and specific plasmid sequence type (ST), as for *bla*_CTX-M-1_/IncI1/ST3 and *bla*_CMY-2_/IncI1/ST12 [66].

A number of distinct transmission routes were related to the transmission of ARB between human and companion animals, such as physical injuries, inhalation, contact with urine, and faecal–oral transmission. The faecal–oral route is of particular concern. For example, in a longitudinal study in the Netherlands, a majority of dogs were found to be intermittent faecal shedders of ESBL-producing *E. coli* and many tested positive for different ESBL genes over time [67]. A high prevalence of bacteria exhibiting resistance to fluoroquinolones (18%) and ESC (18%) was observed in faeces from 269 dogs in veterinary practices in France and Spain, suggesting dogs may form a large reservoir of CTX-M-1 and CMY-2 producers [66]. Canine shedding of *Campylobacter* is also a potential source of zoonotic transmission of, and resistance to, ciprofloxacin and nalidixic acid [68]. This suggests that presence of animal faeces in urban areas offers an additional public-health problem associated with the growing pet and free-roaming animal populations in large cities.

Bacteria isolated from wounds/abscess, ear swabs, and urine of sick pets [69] showed resistance to multiple antibiotic classes and to broad-spectrum antibiotics. Many antimicrobial agents administered to companion animals are similar to those prescribed to humans. Frequent use of broad-spectrum antimicrobials, especially those critically important for human medicine, can result in the transfer of ARB between companion animals and humans. For example, in a study across three European countries (Belgium, Italy, and the Netherlands), the most frequently prescribed antimicrobial to dogs and cats was found to be amoxicillin-clavulanate [70]. Broad-spectrum antimicrobials and critically important antimicrobials for human medicine represented 83% and 71% of the total number of treatments, respectively. Schmidt et al. [71] found that the impact of treatment with β-lactams or fluoroquinolones on third-generation cephalosporin resistance, AmpC-producing, multi-drug resistant, and/or fluoroquinolone-resistant *E. coli* was most acute immediately after treatment, but the effect lessened by the third-month post-treatment [71]. 

Many bacterial strains recovered from dogs and their owners showed phenotypic and genotypic similarities. In 27 households in New Zealand, pet dogs were found to carry the same *E. coli* strain producing ESBL and AmpC β-lactamases as the household members with a urinary tract infection, suggesting likely transmission between humans and animals (or vice versa) within the home environment [72]. Carvalho et al. [73] demonstrated the sharing of multi-drug resistant *E. coli* strains in 9.5% (4/42) of the pairs of isolates from dogs and their owners. *E. coli* ST 405 isolates were found to carry multiple *bla*_CMY-2_ genes on the chromosome and spread between companion animals and humans in South Korea [74]. In a study regarding ESC-resistant *Enterobacteriaceae* in companion animals and humans, the majority of the ESC resistance genes were *bla*_CMY-2-like_ (26.4%), followed by *bla*_CTX-M-55_ (17.2%) and *bla*_CTX-M-14_ (16.1%), whereas *bla*_CTX-M-15_ (28.6%) was predominant in human samples [65]. The prevalence of *sfa*, *hly*, and *cnf* genes in *E. coli* isolated from canine faeces was similar to the owner isolates [75]. Moreover, transmission of MRSA between dogs and their owners has also been reported [76]).

Pet birds are the third most common companion animals after dogs and cats. The majority of caged birds are from two orders: Passeriformes (including canaries and finches) and Psittaciformes (including parrots, parakeets, and lovebirds). Di Francesco et al. [77] evaluated the AMR of Gram-negative species isolated from 456 domestic canaries, showing the presence of multiple resistance, especially against amoxycillin, erythromycin, spiramycin, tiamulin, and tylosin. In another work, various genes encoding ESBL, metallo-β-lactamases, serin-carbapenemases, AmpC β-lactamases, plasmid-mediated quinolone resistance (PMQR) genes, and those conferring resistance against aminoglycosides were detected in *E. coli* isolates from parakeets and parrots in a study carried out on 265 companion birds [78]. Furthermore, in a study involving 735 clinically healthy birds belonging to Fringillidae (*Carduelis carduelis, Serinus canaria*), Estrildidae (*Erythrura gouldiae*, *Lonchura striata domestica, Taeniopygia guttata*), Psittacidae (*Melopsittacus undulatus*, *Agapornis roseicollis*), and Columbidae (*Columba livia domestica*) families, 7.8% of the examined birds were positive for *P. aeruginosa*, with all the strains being resistant to at least one antibiotic and the majority showing multi-drug resistance [79]. 

Other, more exotic, companion animals are also recognised to be reservoirs of different zoonotic microorganisms. For example, cross-infection of multi-drug resistant *P. aeruginosa* between captive snakes and owners has been reported [80,81]. Other authors reported that commensal enteric bacteria from Tokay geckos (*Gekko gecko*) imported through the pet trade displayed resistance against many antibiotics including ampicillin, amoxicillin/clavulanic acid, cefoxitin, chloramphenicol, kanamycin, and tetracycline [82]. Similarly, pet turtles purchased from pet shops and online markets in Korea harboured *P. aeruginosa* strains carrying acquired resistances to imipenem, colistin, streptomycin in addition to intrinsic resistance to other antibiotics [83]. Finally, there are several issues related to exotic animals: for example, their faeces may be released into the sewerage, enter the WWTP and, in some cases, can be released into the environment [84].

Companion animals are clearly a reservoir of antimicrobial resistance and antibiotic resistance genes, suggesting that the antimicrobial prescription for treatment of companion animals needs to be carefully monitored and regulated. Within a One-Health approach, surveillance of both prescriptions for, and resistances within, companion animals should be a priority in the fight against antimicrobial resistance in the urban context.

## 6. Urban Wastewater

Urban wastewater contributes to the spread of ARM because it contains urban sewage [85]. There is an extensive literature about this topic containing many reviews and research papers about the contribution of urban WWTP and sewage to the spread of ARGs and ARB [86,87,88,89,90,91,92]. As highlighted throughout this review, wastewater should be considered one of the most relevant routes linking urban areas (e.g., hospitals, transports, and green spaces) and the environment. However, in the context of this review, one of the most interesting aspects is unintentional leakage from urban sewers (due, for example, to broken pipes) and the subsequent release of contaminated water into the urban environment before appropriate treatment [93,94]. This is quite common in lower- and middle-income countries where water is often supplied through networks that are not constantly pressurized, leading to water stagnation and/or contamination with microbial-polluted water [95]. Moreover, environmental release of contaminated water can occur even in the case of functional networks, as during combined sewer overflow events. Under these circumstances, a higher water flow, due, for example, to large rainstorms, exceeds the capacity of the WWTP, leading to the release of untreated water that can pollute drinking water supplies with microbial contaminants, including those of faecal origin [96,97].

Regular investigations regarding the microbial community of drinking water can help local authorities take proper measures against the diffusion of ARGs and ARB, as recently shown [98,99]: for example, metagenomic analyses can be used to determine the diversity and the structure of bacterial communities in drinking water, and culture-based approaches can allow the evaluation of antibiotic resistance and the ability of isolated bacteria to form biofilms in drinking water.

## 7. The Urban Network of ARB/ARGs

The urban context is an extremely complex network of interaction among infrastructures, humans, and animals (Figure 1 and Figure 4 and Table 1).

All these interactions can transfer ARGs and ARB via direct (i.e., human-to-human) and indirect contacts (i.e., object-mediated). According to the UN Department of Economic and Social Affairs (UN DEAS), the proportion of the world’s population living in urban areas will increase from 55% to 68% by 2050 [113]. This increase will multiply such interactions within the next decades, posing further risks. 

The gene surveillance of each environment (depending on the risk) is important to control possible spread of infections. However, using resistome profiling to assess risk is complicated, for at least three reasons. The first is that an accurate resistome analysis requires previous knowledge of the target genes. New resistance genes (generated, for example, through recent acquired mutations) cannot be detected by the analysis and, currently, software used for function prediction by sequence similarity has a limited utility in risk assessment [114]. Second, it is important to understand the role of mobile genetic elements, the functional-taxonomy (annotation of ARGs and taxa) and pipelines that can help to characterize targeted ARGs and their association with mobile genetic elements [15,115,116]. Studies performed to simulate the release of different concentrations of antimicrobial in the environment may provide useful information about the fate of ARB/ARGs in specific environmental contexts [117,118]. Third, ARGs evolve and replicate. Reduction in ARGs (or ARB), for example, due to wastewater treatments, does not assure that ARB will not (re)appear and will not be able to restore a new replicative population. Therefore, there is no proportionality between abundance of AMRs and risk [5,119,120]. 

The current risk assessment for ARGs/ARB is based on four domains; hazard identification, release assessment, exposure assessment, and consequences [121,122]. An additional domain should be added: it should consider the possibility of evolution and replication, taking into account also that persistence and replication of ARGs/ARB are not only driven by selective pressure. 

Equally, there are issues regarding the wider urban context; relationships between different urban centres, responses of their individual resistomes to increased intercity/international transport and their influences upon non-urban resistomes. There may be some parallels with the ideas associated with the Island Biogeography concept [123]. Urban resistomes are dynamic, highly selective environments, and they are equivalent to ‘mainland’ biomes. As transport increases, to and from urban and non-urban environments, natural resistomes supporting indigenous ARGs/ARB diversity will face an increasing import of globally occurring urban ARGs/ARB, moderated only by the relative fitness of immigrant and indigenous bacteria [124] and by innate local microbiome resistance [125,126]. Barberán et al. [127] suggest that local microbiome composition results from indigenous speciation and extinction plus colonization by, and dispersal to, a global microorganism pool. As urbanization intensifies it is likely that urban resistome elements will tend to homogenise and to dominate the ‘global pool’. It may be that, under increasing urban influence, all resistomes will tend to homogenise with the consequences of natural biodiversity loss, ecological malfunction [128,129,130], and a refractory global antibiotic resistance crisis.

## Figures and Tables

**Figure 1 antibiotics-11-00512-f001:**
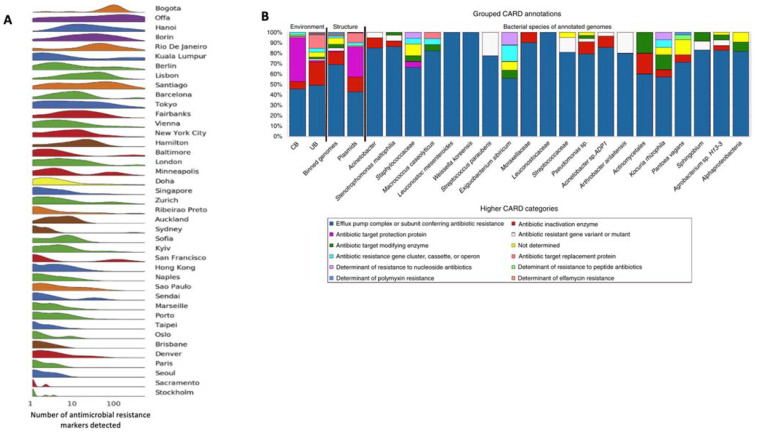
(**A**) Number of detected antimicrobial resistance markers by city. Colour represents different regions. Clusters of high antimicrobial resistance markers diversity were not evenly distributed across cities. (**B**) The portion of the Comprehensive Antibiotic Resistance Database (CARD) and antibiotic classes in controlled built environments (CB, microbial confinement and cleaning operations) and naturally unrestricted buildings (UB, houses with a high level of influence from the surrounding outdoor environment). The abundance chart also shows binned genomes and plasmids and for individual binned genomes referring to individual species. Figure modified from [11] (panel A) and [12] (panel B).

**Figure 2 antibiotics-11-00512-f002:**
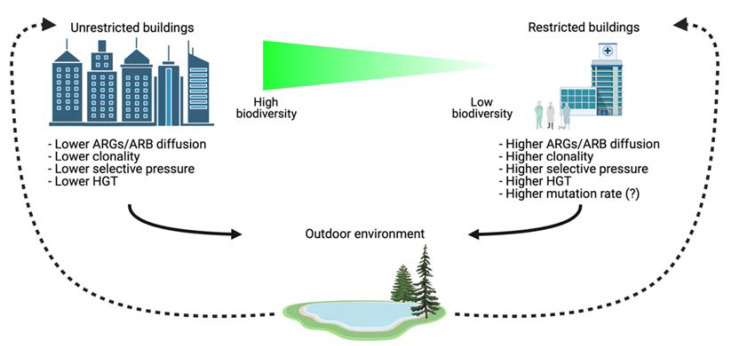
Differences between restricted and unrestricted buildings and suggested routes connecting outdoor and indoor settings. Restricted buildings have lower biodiversity when compared with unrestricted buildings. ARGs and ARB present in outdoor environments (dotted lines, high microbial biodiversity) can be transferred into indoor settings (with lower microbial biodiversity), where more resistant forms can be selected through, for example, use of antibiotics. Eventually, these selected and resistant ARGs and ARB can be released outdoors (continuous lines) by different vectors, such as people, pets, and wastewater. In the environment selected ARGs/ARB can find new ways for recombination due to higher biodiversity and integration of ARGs in the environmental microbial communities.

**Figure 3 antibiotics-11-00512-f003:**
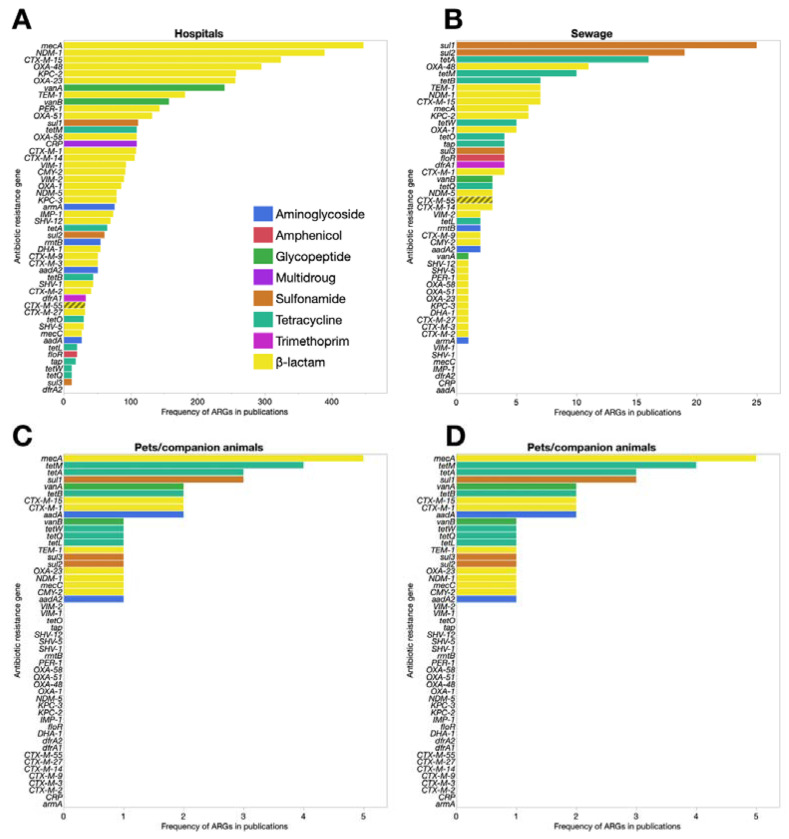
Frequency of papers regarding the top 50 ARG subtypes from 1990. The panels from (**A**–**D**) represent ARGs and their respective antibiotic families in different urban contexts. The number of manuscripts identifying the genes were extracted from PubMed [17].

**Figure 4 antibiotics-11-00512-f004:**
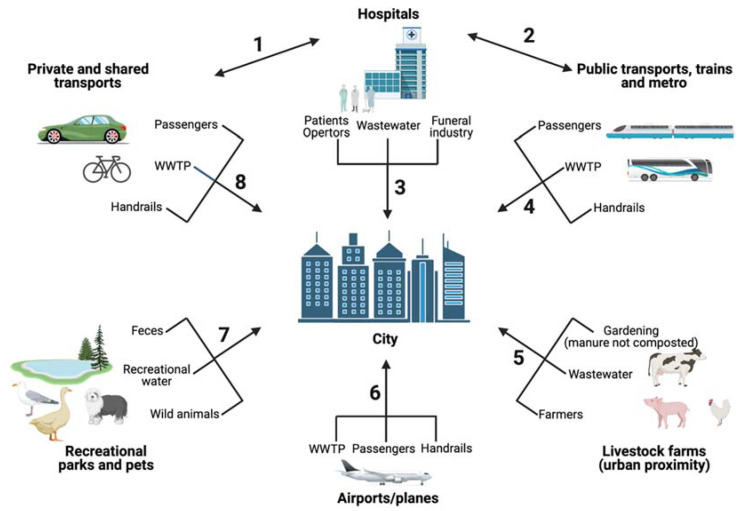
ARGs/ARB contamination routes within the urban environment. Each route discussed in this work is identified with a number and the corresponding supporting references are listed in Table 1. This figure has been prepared using resources from Freepik.com. WWTP: wastewater treatment plants.

**Table 1 antibiotics-11-00512-t001:** List of references regarding ARGs/ARB contamination routes depicted in Figure 4.

Route ^1^	References
1	[43]
2	[35,36]
3	[12,20,21,22,25,26,28,29]
4	[34,37,38,39,40]
5	[100,101,102,103,104,105,106,107,108,109,110,111,112]
6	[30,31,32,33]
7	[10,20,63,65,66,67,68,69,72,73,74,75,51,76,77,78,79,80,81,82,83,84,52,53,54,55,56,61,62]
8	[41,42,43,44]

^1^ The numbers refer to the routes in Figure 4.

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
