# Peer review of "The Bacterial Urban Resistome: Recent Advances"

_antibiotics, 2022, doi:10.3390/antibiotics11040512_

Round 1

Reviewer 1 Report

This review is an interesting and well-written article focused on the bacterial urban resistome, which includes some aspects within a One-health approach. Some minor considerations:

Figure 3: Resistance due to mecA and mecC affects to betalactams, so both genes should be represented in yellow.

Line 410-411: Pseudomonas aeruginosa is intrinsically resistant to amoxicillin, cephalotin, cefoxitin, cotrimoxazole, chloramphenicol and nalidixic acid. Only the resistance to imipenem, colistin and streptomycin is acquired.

Figure 4. Please describe what WWTP means at the bottom of the figure.

Although the review is well-structured in six paragraphs, it should perhaps be desirable to add another on urban wastewater, in order to briefly comment on the importance of this reservoir and the usefulness of its analysis for  continous global surveillance (doi: 10.1038/s41467-019-08853-3 )

Author Response

We thank the reviewer for the positive feedback. We have replied point-by-point to all comments by using a blue font. Additions are highlighted in blue in the manuscript.

Reviewer 1

This review is an interesting and well-written article focused on the bacterial urban resistome, which includes some aspects within a One-health approach. Some minor considerations:

Figure 3: Resistance due to mecA and mecC affects to betalactams, so both genes should be represented in yellow.

We thank the reviewer for finding this graphical error. We have corrected the bars’ color. Figure 3, line 108.

Line 410-411: Pseudomonas aeruginosa is intrinsically resistant to amoxicillin, cephalotin, cefoxitin, cotrimoxazole, chloramphenicol and nalidixic acid. Only the resistance to imipenem, colistin and streptomycin is acquired.

We have simplified the sentence: “Similarly, pet turtles purchased from pet shops and online markets in Korea harbored P. aeruginosa strains carrying acquired resistances to imipenem, colistin, streptomycin in addition to intrinsic resistance to other antibiotics”. Line 419

Figure 4. Please describe what WWTP means at the bottom of the figure.

 We added in the caption of Figure 4. Line 458.

Although the review is well-structured in six paragraphs, it should perhaps be desirable to add another on urban wastewater, in order to briefly comment on the importance of this reservoir and the usefulness of its analysis for  continous global surveillance (doi: 10.1038/s41467-019-08853-3 )

We thank the reviewer for this suggestion. We have added a new paragraph about urban wastewater and in particular on the possible routes leading to the contamination of drinking water supplies. Lines 429-451.

Reviewer 2 Report

Review of antibiotics-1673646

This is a nice review about antibiotic-resistant genes and bacteria (ARGs, ARB), as well as the carrier, places, and their routes to be in contact with humans. This review discusses not just the common or mainstream routes for ARGs or ARB transmission, but also the new or less-explored routes such as the funeral, or less-common “reservoir” such as pet birds and reptiles (snakes, geckos, turtles). This will be a solid contribution to the knowledge of antibiotics. There are some issues to be addressed before the publication, as follows:

  1. The title is quite short. How about the addition of “.. an updated review”, or “Recent advances of…”, or “..from 2000-2022” ?
  2. Line 117: Please write “hospitals” with lowercase h.
  3. Please incorporate these recent articles related to the bacterial profile in drinking water in urban places:
  • Water 2022, 14(6), 908; https://doi.org/10.3390/w14060908
  • Water 2022, 14(6), 944; https://doi.org/10.3390/w14060944

  1. Line 298-325: Please separate this long paragraph as 2-3 shorter paragraphs.
  2. Line 432-470: Please separate this long paragraph as 3-5 shorter paragraphs.
  3. Please write the complete list of authors, do not use “et al” for these reference numbers: 11, 15, 16, 20, 23, 31, 34, 35, 41, 50, 66, 70, 72, 78, 80,
  4. Please write all scientific names in italic, with uppercase letter for the genus, in these reference numbers: 24, 37, 56, 68, 71, 72, 73, 74, 75, 76 (Uppercase S), 77, 78, 79, 80, 81, 82,83, 89, and 90

Author Response

We thank the reviewer for the positive feedback. We have replied point-by-point to all comments by using a blue font. Additions are highlighted in blue in the manuscript.

Reviewer 2

This is a nice review about antibiotic-resistant genes and bacteria (ARGs, ARB), as well as the carrier, places, and their routes to be in contact with humans. This review discusses not just the common or mainstream routes for ARGs or ARB transmission, but also the new or less-explored routes such as the funeral, or less-common “reservoir” such as pet birds and reptiles (snakes, geckos, turtles). This will be a solid contribution to the knowledge of antibiotics. There are some issues to be addressed before the publication, as follows:

The title is quite short. How about the addition of “.. an updated review”, or “Recent advances of…”, or “..from 2000-2022” ?

We rephrased the title with: The bacterial urban resistome: recent advances

Line 117: Please write “hospitals” with lowercase h.

done

Please incorporate these recent articles related to the bacterial profile in drinking water in urban places:

  • Water 2022, 14(6), 908; https://doi.org/10.3390/w14060908
  • Water 2022, 14(6), 944; https://doi.org/10.3390/w14060944

We thank the reviewer for this suggestion. We have added a new paragraph about urban wastewater and on the possible routes leading to the contamination of drinking water supplies. Lines 429-451.

Line 298-325: Please separate this long paragraph as 2-3 shorter paragraphs.

We separated the paragraph as requested. Lines 301-329.

Line 432-470: Please separate this long paragraph as 3-5 shorter paragraphs.

We separated the paragraph as requested. Lines 463-504.

Please write the complete list of authors, do not use “et al” for these reference numbers: 11, 15, 16, 20, 23, 31, 34, 35, 41, 50, 66, 70, 72, 78, 80,

To our understanding the style of the bibliography is formatted in agreement with the requirements of the journal, that suggests including only the first ten authors of the cited work.

Please write all scientific names in italic, with uppercase letter for the genus, in these reference numbers: 24, 37, 56, 68, 71, 72, 73, 74, 75, 76 (Uppercase S), 77, 78, 79, 80, 81, 82,83, 89, and 90

We apologize for this inconvenient: these mistakes were due to the use of a bibliography software. Scientific names have been formatted as request in the new version of the manuscript.